# Modified Desolvation Method Enables Simple One-Step Synthesis of Gelatin Nanoparticles from Different Gelatin Types with Any Bloom Values

**DOI:** 10.3390/pharmaceutics13101537

**Published:** 2021-09-22

**Authors:** Pavel Khramtsov, Oksana Burdina, Sergey Lazarev, Anastasia Novokshonova, Maria Bochkova, Valeria Timganova, Dmitriy Kiselkov, Artem Minin, Svetlana Zamorina, Mikhail Rayev

**Affiliations:** 1Perm Federal Research Center of the Ural Branch of The Russian Academy of Sciences, Lab of Ecological Immunology, Institute of Ecology and Genetics of Microorganisms, 614081 Perm, Russia; krasnykh-m@mail.ru (M.B.); timganovavp@gmail.com (V.T.); mantissa7@mail.ru (S.Z.); mraev@iegm.ru (M.R.); 2Department of Biology, Perm State University, 614068 Perm, Russia; izotrofan@mail.ru (O.B.); lasest1999@gmail.com (S.L.); anast218bio@gmail.com (A.N.); 3Center for Immunology and Cellular Biotechnology, Immanuel Kant Baltic Federal University, 236016 Kaliningrad, Russia; 4Perm Federal Research Center of the Ural Branch of The Russian Academy of Sciences, Institute of Technical Chemistry, 614013 Perm, Russia; dkiselkov@yandex.ru; 5Lab of Applied Magnetism, M.N. Mikheev Institute of Metal Physics of the UB RAS, 620108 Yekaterinburg, Russia; calamatica@gmail.com; 6Faculty of Biology and Fundamental Medicine, Ural Federal University Named after The First President of Russia B.N. Yeltsin, 620002 Yekaterinburg, Russia

**Keywords:** encapsulation, nanoprecipitation, coacervation, manufacturing, yield, nanocarriers, drug delivery

## Abstract

Gelatin nanoparticles found numerous applications in drug delivery, bioimaging, immunotherapy, and vaccine development as well as in biotechnology and food science. Synthesis of gelatin nanoparticles is usually made by a two-step desolvation method, which, despite providing stable and homogeneous nanoparticles, has many limitations, namely complex procedure, low yields, and poor reproducibility of the first desolvation step. Herein, we present a modified one-step desolvation method, which enables the quick, simple, and reproducible synthesis of gelatin nanoparticles. Using the proposed method one can prepare gelatin nanoparticles from any type of gelatin with any bloom number, even with the lowest ones, which remains unattainable for the traditional two-step technique. The method relies on quick one-time addition of poor solvent (preferably isopropyl alcohol) to gelatin solution in the absence of stirring. We applied the modified desolvation method to synthesize nanoparticles from porcine, bovine, and fish gelatin with bloom values from 62 to 225 on the hundreds-of-milligram scale. Synthesized nanoparticles had average diameters between 130 and 190 nm and narrow size distribution. Yields of synthesis were 62–82% and can be further increased. Gelatin nanoparticles have good colloidal stability and withstand autoclaving. Moreover, they were non-toxic to human immune cells.

## 1. Introduction

Gelatin is a product of partial hydrolysis of collagen. In the course of gelatin preparation, collagen is pre-treated under acidic or alkaline conditions, which results in obtaining two types of gelatin: type A and type B, respectively. The main sources of gelatin are bovine skin, bovine hides, and cattle and pork bones, whereas fish and poultry gelatins are used to a limited extent [1]. Gelatin from cold-water fish contains a lower percentage of proline and hydroxyproline which are involved in the formation of collagen-like triple helices and therefore has inferior gelation properties in comparison with mammalian gelatins [1,2]. Being biocompatible (included in FDA’s GRAS list), low-immunogenic, cheap, and commonly available biopolymer gelatin gains popularity in biomedicine, biotechnology, and food science [3]. Conditions in which hydrolysis of collagen is performed affect the size distribution of resulting gelatin molecules. Size distribution of gelatin molecules usually correlates with gel strength which is expressed as a bloom value: the longer the gelatin polypeptide chains the higher the gel strength and bloom value [4].

Gelatin, as well as many other proteins, is used in the form of gelatin nanoparticles. Nanoparticles are small particles with sizes from several nanometers to hundreds of nanometers [5]. Gelatin nanoparticles consist of numerous gelatin molecules which are covalently cross-linked or stabilized by a coating layer [3]. Drug delivery is arguably the scientific field most intensively taking advantage of gelatin nanoparticles. Various drug molecules, DNA, imaging agents (e.g., radiotracers or fluorescent dyes), or nutrients can be physically entrapped in the nanoparticle body or chemically attached to their surface [6]. Tuning of nanoparticles’ diameter, coating, and cross-linking degree allows to control circulation time, cellular uptake, as well as drug release rate [6,7]. Gelatin nanoparticles offer a number of unique advantages, namely, they provide improved pharmacokinetics, release profile, and localized delivery of drugs while retaining favorable properties of gelatin including biodegradability along with low cytotoxicity and low immunogenicity [7]. Many reports of gelatin-based nanotherapeutics and nanovaccines were made in the past years [8]. Below some representative examples of successful in vivo applications of gelatin nanoparticles-based nanomedicines are presented.

Application of gelatin nanoparticles allowed the same therapeutic effect with the five-fold lower dose of timolol maleate for glaucoma treatment on a mouse model in comparison with conventional therapy (free timolol maleate) [9]. Gelatin nanoparticles and their aminated counterparts exhibited immunomodulatory efficiency comparable to that of aluminum adjuvants being non-immunogenic by themselves [10]. Pegylated gelatin nanoparticles showed excellent biocompatibility and significantly improved release kinetics and bioavailability of ibuprofen after parenteral administration [11]. Gelatin nanoparticles loaded with immunostimulatory cytosine-phosphate-guanosine oligodeoxynucleotides provided long-term positive effects in horses with asthma and showed higher efficacy in comparison with standard therapy [12]. Antimicrobial gelatin nanoparticles modified with selenium nanoparticles and ruthenium complexes and coated with erythrocyte membranes were tested in vivo on mice. Nanoparticles accumulated in the injury site and provided elimination of methicillin-resistant Staphylococcus aureus; their efficiency was equal to that of vancomycin [13].

Potential applications of gelatin nanoparticles are not limited to therapeutics and vaccine development. Gelatin nanoparticles and microparticles together with molecular gelatin can serve as cheap collagen substitutes imitating extracellular matrix in cell culturing and tissue engineering [14]. Gelatin nanoparticles improve the mechanical properties of bioinks for 3D bioprinting [15,16] and increase circulating tumor cell capture in a microfluidic device [17]. Preparation of Pickering emulsions for food chemistry is another prominent application of gelatin nanoparticles [18].

Desolvation is one of the most popular techniques for gelatin nanoparticle synthesis [3]. This method relies on the addition of poor solvents (usually acetone, alcohols, or acetonitrile) to the aqueous protein solution. Desolvation of gelatin is regularly performed in two steps according to the method described by Coester et al. [19] and further optimized by researchers from the same scientific group [20,21]. The first step of desolvation includes the addition of non-solvent to gelatin solution resulting in sedimentation of high-molecular gelatin fractions. Sediment is dissolved in water, then, after the pH adjustment, repeated addition of non-solvent results in the formation of gelatin nanoparticles. Being a relatively simple and accessible method of synthesis of stable and biocompatible gelatin nanoparticles, two-step desolvation is widely used in various fields. Shortcomings of two-step desolvation are low particle yields, lack of reproducibility of the first desolvation step, and difficult process scale-up [22]. Therefore, numerous efforts were made to develop a more straightforward, one-step technique. It has been shown that the presence of low-molecular-weight fractions (more than 20% of fractions with molecular weight less than 65 kDa) in gelatin preparations leads to the formation of non-stable and polydisperse nanoparticles. These very fractions need to be removed with the first desolvation step [20,21].

Several approaches were proposed to prepare gelatin nanoparticles by the desolvation method in one step. The first approach is to use custom-made or recombinant gelatin lacking low-molecular-weight fractions [21,23]. The disadvantage of this method is limited availability and the high cost of starting material. Commercially available high-bloom gelatin (bloom value of 300) allows to skip the first desolvation step [22], however, resulting nanoparticles tend to aggregate [24]. Vacuum filtration was used to get rid of large molecular weight gelatin and increase the homogeneity of gelatin before desolvation [25]. Shamarekh et al., prepared gelatin enriched with high-molecular-weight fractions from commercial gelatin and used it as a starting material [26]. Despite these last two methods allowing desolvation to be made in one step, they are rather quasi-one-step than true one-step because both of them still require depletion of smaller gelatin molecules.

Surprisingly, several research groups reported the synthesis of uniform gelatin nanoparticles by one-step desolvation without removal of low-molecular-weight fractions [9,27,28,29]. All these groups used gelatin with bloom 225 or lower, which is expected to be not compatible with the one-step method. Most of these works lack an explanation of why proposed synthesis protocols are effective, however, Ofokansi et al., claimed that neutral pH facilitated stability and homogeneity of nanoparticles [29].

We revealed that the stirring speed of the gelatin solution dramatically influences the desolvation process. Intensive stirring promotes gelatin aggregation. Using a simple one-step stirring-free approach we previously prepared stable and homogeneous gelatin nanoparticles from gelatin with bloom values as low as 75 [30]. Based on these findings we intended to develop a facile method for nanoparticle preparation from any type of gelatin with any bloom number. Hence, the goals of this work were as follows:To confirm the effect of stirring on desolvation of gelatin;To study the influence of gelatin pH, concentration, and non-solvent type on the size and yield of gelatin nanoparticles;Using optimized conditions to synthesize nanoparticles from porcine, bovine, and fish gelatin with different bloom values (including lowest values available) in a hundreds-of-microgram scale;To study storage stability and colloidal stability of resulting nanoparticles;To load model hydrophobic molecules into gelatin nanoparticles;To assess the effect of sterilization on the integrity of gelatin nanoparticles;To study cytotoxicity of gelatin nanoparticles prepared by modified desolvation method.

## 2. Materials and Methods

### 2.1. Materials

Gelatin B, 75 bloom (lot# G6650); gelatin B, 225 bloom (lot# G9382); cold-water fish gelatin (lot# G7041), gelatin A, 62 bloom (lot# 48720); gelatin A, 180 bloom (lot# 48722) BCA assay kit, 1,10-phenanthroline, and boric acid were obtained from Sigma Aldrich (Burlington, MA, USA). Glutaraldehyde (50%) was obtained from ITW Reagents (Glenview, IL, USA). Trypsin was obtained from Samson-Med (Saint-Petersburg, Russia). Diacoll was obtained from Dia-M (Moscow, Russia). Propidium iodide was obtained from eBioscience (Santa Clara, CA, USA). DMSO was obtained from Tula Pharmaceutical Plant (Tula, Russia). Water for injections was obtained from Solopharm (Saint-Petersburg, Russia). Sodium hydroxide, sodium chloride, sodium hydrogen phosphate, sodium dihydrogen phosphate, sodium bicarbonate, sodium carbonate, glycine were obtained from ITW Reagents (Glenview, IL, USA). Isopropyl alcohol, ethanol, methanol, hydrochloric acid, acetic acid were obtained from Vekton (Saint-Petersburg, Russia). 

4-(4-methylphenyl)-2,4-dioxobutanoic acid was obtained from commercially available reagents by the Claisen condensation [31] and kindly provided by Dr. Ekaterina Khramtsova, department of Organic Chemistry, PSU.

The following instrumentation was used: peristaltic pump, LKB (Upsalla, Sweden), Synergy H1 plate reader, BioTek (Winooski, VT, USA), Multiskan Sky UV-Vis Reader, Thermo (Waltham, MA, USA) and ZetaSizer NanoZS particle analyzer, Malvern (Malvern, UK), CytoFLEX flow cytometer, Beckman Coulter (Brea, CA, USA), SV-10 viscometer, A&D (Tokyo, Japan). Multipette M4, Eppendorf (Hamburg, Germany) was used for accurate dispensing of viscous gelatin solutions. 

### 2.2. Preparation of Gelatin Stock Solutions

Gelatin powder was added to a certain volume of water and incubated at +40 °C until a clear solution was obtained. Gelatin solution was aliquoted and stored at +4 °C. The concentration of gelatin was determined gravimetrically as follows. Gelatin solution (1 mL) was added to the porcelain crucible and dried to constant weight at subsequently +95 and +140 °C. Three replicates were done for each sample. The concentration of gelatin nanoparticles was measured in the same way.

### 2.3. Synthesis of Gelatin Nanoparticles in Hundred-of-Milligram Scale

Gelatin A (bloom 62 and 180), gelatin B (bloom 75 and 225), and fish gelatin were diluted in water to 10 mg/mL, then pH was adjusted to 10 (to 11 for gelatin A and fish gelatin) with 1 M NaOH. One hundred milliliters of the resulting solution were desolvated by 500 mL of isopropyl alcohol, and the mixture was incubated for 30 min at +37 °C in the water bath. Gelatin solutions and isopropyl alcohol were kept in the water bath at +37 °C prior to mixing. Then, 22.5 mL of 0.8% glutaraldehyde was quickly added, followed by 30 min long incubation at +37 °C in the water bath. Cross-linked nanoparticles were transferred into polycarbonate 85 mL centrifuge tubes and centrifuged at 15,000 *g* for 60 min. Pellets were combined and redispersed in 60 mL of water with sonication (10–30 s, 60% amplification, 3 mm probe, approx. 8 W), resulting suspensions were centrifuged at 15,000 *g* for 30 min two more times. After the final centrifugation, 40 mL of water was added to the pellet, and the resulting suspension was sonicated for 20 min (60% amplification, 6 mm probe, approx. 25 W) on ice. The concentration of nanoparticles was determined by gravimetric analysis.

For fluorescence measurements samples were diluted to 1 mg/mL with water; then, 100 μL of each sample was transferred into the wells of black 96-well plates. The size of nanoparticles was determined by the dynamic light scattering (DLS) technique. For DLS measurements nanoparticles were diluted at 1:375 in water. Hereinafter z-average hydrodynamic diameters (Dh) are given. In order to obtain scanning electron microscopy (SEM) images, nanoparticles were diluted in water to 1 μg/mL, dropped at 5 × 5 mm silicon wafer, and dried overnight at room temperature. For SEM experiments glycine-quenched nanoparticles were taken to reduce the possibility of interaction between free aldehyde groups and amine groups located on the nanoparticles’ surface in the course of drying. 

### 2.4. Preparation of Gelatin Nanoparticles Loaded with Fluorescent Europium Chelates

Gelatin A (bloom 62 and 180), gelatin B (bloom 75 and 225), and fish gelatin were diluted in water to 10 mg/mL then pH was adjusted to 10 (to 11 for gelatin A and fish gelatin) with 1 M NaOH. Four milliliters of the resulting solution were desolvated by 20 mL of ethanol containing 4-(4-Methylphenyl)-2,4-dioxobutanoic acid, 1,10-phenanthroline, and europium chloride (concentrations were 180, 60, and 60 μM, respectively), and the mixture was incubated for 30 min at +37 °C in the thermostat [32]. Solutions containing gelatin and fluorescent complexes were kept on the water bath at +37 °C before mixing. Then, 900 μL of 0.8% glutaraldehyde was added, followed by 30 min long incubation at +37 °C. Nanoparticles were transferred into polycarbonate 85 mL centrifuge tubes and centrifuged at 15,000 *g* for 60 min. Pellet was redispersed in 4 mL of water with sonication and centrifuged three times at 20,000 *g* for 30 min. After each wash pellet was redispersed in water by sonication (10–30 s, 60% amplification, 3 mm probe, approx. 8 W). Supernatants obtained after the final washing step were collected. For fluorescence measurements 10fold dilutions of nanoparticles in water were prepared; then 100 μL of each dilution were transferred into the wells of black 96-well plates.

### 2.5. Steam Autoclaving

Glycine quenching. Before sterilization, 1 M glycine-NaOH buffer pH 9.3 was added to gelatin nanoparticles suspension (1 part of buffer per 9 parts of suspension) and the resulting mixture was incubated for 1 h at +37 °C on a rotator (10 rpm, 360 degrees). Nanoparticles were then washed three times with water by centrifugation at 15,000 *g* for 1 h. The concentration of nanoparticles was determined by gravimetric analysis. 

Autoclaving. Five milliliters of the resulting nanoparticle suspension were placed in the 15 mL amber glass vials and autoclaved for 15 min at 0.5 atm above atmospheric pressure. Suspensions were cooled at room temperature and stored at +4 °C. Control nanoparticles were kept at +4 °C.

Removal of nanoparticle aggregates after autoclaving. A total of 1 mL of nanoparticle suspension was moved to centrifuge tubes. Nanoparticles were centrifuged at 1000 *g* for 10 min. After centrifugation nanoparticle size was measured by DLS. Triple replicates were performed for each measurement.

Characterization. The optical density of nanoparticle suspensions was measured before and after centrifugation. The suspension was diluted in distilled water. The measurement was performed at 600 nm. For DLS measurements nanoparticles were diluted at 1:375 in PBS (pH 7). Zeta potential of autoclaved nanoparticles was measured at pH 7 and ionic strength of 0.06 M. Ionic strength was adjusted with 1 M KNO3. The measurements were done with three technical replicates.

### 2.6. Assessment of Nanoparticle Stability at Different pH and High Salt Concentrations

Stability of nanoparticles at different pH values. Nanoparticles were diluted to 50 μg/mL in buffer solutions with pH ranging from 4 to 10. The following buffers were used: 10 mM acetate buffer, pH 4 and 5; 10 mM sodium phosphate buffer, pH 6, 7, and 8; 10 mM borate buffer, pH 9 and 10. Ionic strength was adjusted to 0.15 M by the addition of NaCl. Nanoparticle size at each pH was measured immediately by DLS. Three replicates were done for each measurement. Nanoparticle suspensions were stored for 7 days in plastic cuvettes, which were placed in a wet chamber. On days 1 and 7 additional size measurements were performed.

Stability of nanoparticles at different salt concentrations. Nanoparticles were diluted to 50 μg/mL in a phosphate buffer, pH 7. The ionic strength of the solution was increased to 0.5, 1, 2, and 3 M by the addition of NaCl. Nanoparticle size was measured by DLS for each salt concentration. Viscosity of NaCl solutions was determined with the aid of viscometer and were 1.265 mPa·s (3 M NaCl), 1.145 mPa·s (2 M NaCl), 0.992 mPa·s (1 M NaCl) and 0.983 mPa·s (0.5 M NaCl).

Measurement of nanoparticle zeta potential at different pH. Nanoparticle suspensions were diluted to 50 μg/mL in the following buffers: 10 mM acetate buffer, pH 4 and 5; 10 mM sodium phosphate buffer, pH 6, 7, and 8; 10 mM borate buffer, pH 9 and 10. Ionic strength was adjusted to 0.06 M by the addition of KNO3. Three technical replicates were performed for each measurement.

### 2.7. Cell Viability Study

Venous blood was drawn from three healthy volunteers (from 23 to 31 years old) into heparin-contained vacuum tubes. Peripheral blood mononuclear cells (PBMC) were isolated from blood plasma by density gradient centrifugation with Diacoll (1077 g/L, Dia-M, Moscow, Russia) at 400 g for 40 min. Isolated cells were washed with Hanks′ Balanced Salt solution three times; then, cells were seeded in duplicates into 96-well plates (200 μL per well, 1 × 10^6^ cells/mL). Thirty microliters of sterilized (see Section 2.5) gelatin nanoparticles diluted in water for injections (WFI) were added into each well. Final concentrations of gelatin nanoparticles were 1000, 250, 62.5, 15.6, and 3.9 μg/mL. The negative and positive controls were WFI and 15% DMSO, respectively [33]. Cells were incubated for 24 h in a humidified atmosphere in the CO_2_ incubator (5% of CO_2_, +37 °C), stained with propidium iodide (PI) (1 μg/mL, 5 μL for 100 μL of cell suspension) for one minute, and analyzed by flow cytometry. The percentage of PI− (living) cells was determined for each sample. 

Monocytes engulfing particles fluoresce in the emission spectrum of propidium iodide (maximum about 615 nm) (Appendix A). Therefore, gates for living (PI−) and dead (PI+) cells were set according to unstained samples and positive/negative controls (Appendix A).

The granularity of nanoparticles-engulfing cells, and, accordingly, the side light scatter (SSC) parameters increases [34]. Therefore, the engulfing of particles by monocytes was determined by the geometric mean of side scattering intensity (Appendix A).

## 3. Results and Discussion

### 3.1. Stirring Promotes Nanoparticle Aggregation in the Course of Desolvation

We revealed that quick one-time addition of non-solvent to aqueous gelatin solution without agitation leads to the formation of monodisperse gelatin nanoparticles. Moreover, in the course of preliminary experiments nanoparticles were successfully prepared from gelatin B with bloom values as low as 75 (molecular weight in the range between 20 and 25 kDa according to manufacturer). This result contradicts conclusions made by other researchers: usually, removal of low-molecular-weight gelatin fractions is necessary to obtain stable and fine nanoparticle suspensions [20,21]. We suggest that stirring of gelatin solution upon addition of non-solvent promotes aggregation of gelatin molecules and, thus, can be completely omitted. 

To prove our suggestion, we performed the following experiment. Ethanol was quickly added to gelatin B (75 bloom) solutions (10 and 20 mg/mL). Then, the suspensions were mixed using three regimes: (1) gentle mixing on a rotator (10 rpm); (2) slow vortexing (100 rpm); (3) fast vortexing (2500 rpm). Three individual batches were prepared for each condition. Their size and polydispersity as well as turbidity (absorbance at 600 nm) were measured immediately after the synthesis. 

At the gelatin concentration of 10 mg/mL, vortexing has little effect and only the highest speed provokes slight growth of size and turbidity. However, vortexing had a dramatic impact when gelatin concentration was doubled: almost 50% growth of mean size and turbidity at a low speed and severe aggregation at a high speed. At the same time, homogeneous suspensions of gelatin nanoparticles with polydispersity indices lower than 0.1 were formed when mixing was performed on the rotator (Figure 1). We did not perform experiments with higher gelatin concentrations, but recently we successfully prepared gelatin nanoparticles using 30 mg/mL gelatin solution which was desolvated under short gentle mixing [30].

Obtained results explain why some researchers were able to carry out the one-step synthesis of gelatin nanoparticles without removal of low-weight gelatin fractions [9,27,28,29,35,36,37]. They desolvated solutions with low gelatin concentrations (1% or less) which are not affected by stirring. 

At the same time, Geh et al., synthesized monodisperse gelatin nanoparticles from 40 and 50 mg/mL gelatin solutions by adding acetone under stirring [22]. The authors used commercially available gelatins A and B with bloom values of 300 and mean molecular weights in the range between 400 and 500 kDa. Therefore, we suggest that low-molecular-weight fractions in gelatin preparations can promote aggregation in the course of desolvation at high total gelatin concentrations (circa 20 mg/mL and more) when stirring is carried out. A small percentage or absence of low-molecular-weight fractions enables nanoparticle synthesis under stirring. To reinforce previous findings and demonstrate the role of stirring and low-molecular-weight gelatin fractions in the desolvation process two more experiments were performed.

Firstly, we desolvated a solution of low-bloom gelatin B (30 mg/mL, pH 9) with ethanol under vigorous stirring and without stirring. One-time addition of non-solvent led to homogeneous suspension of nanoparticles whereas stirring-assisted dropwise addition of ethanol resulted in the formation of gelatin bulk at the bottom of the vial (Figure 2). 

In the second experiment, we desolvated with isopropyl alcohol solutions of gelatins A with bloom values of 300 and 62 (both 30 mg/mL, pH 10) or their 3:1, 2:2, or 1:3 mixtures. The total volume of the gelatin solution was 4 mL. Volume fraction of gelatin A, 62 bloom varied from 0% (4 mL of gelatin A, 300 bloom + 0 mL of gelatin A, 62 bloom) to 100% (0 mL of gelatin A, 300 bloom + 4 mL of gelatin A, 62 bloom). As we said before, Geh and colleagues successfully desolvated [22] solution of gelatin A, 300 bloom under stirring. Therefore, this sort of gelatin is suitable for one-step desolvation. Conversely, gelatin A, 62 bloom has one of the lowest bloom values from commercially available gelatins and should contain mostly low-molecular fractions, being therefore incompatible with the one-step method. By increasing the percentage of low-bloom gelatin in the mixture of two gelatins we studied the role of low-molecular fractions in the desolvation process.

As expected, stirring-assisted desolvation of gelatin mixtures containing 75% and 100% of gelatin A, 62 bloom resulted in sedimentation of sticky gelatin mass at the bottom of the vials (Appendix A). On the contrary, no sign of such aggregation was observed in other vials. Visual inspection revealed that more turbid suspensions were obtained when the percentage of low-bloom gelatin exceeded 25% (Appendix A).

After that, we performed desolvation of low-bloom gelatin A solution under stirring and without stirring, as was done previously for gelatin B, 75 bloom. Again, the homogeneous colloidal solution was obtained in the stirring-free conditions, whereas stirring-assisted desolvation resulted in turbid suspension, containing visible aggregates and gelatin mass at the bottom of the reaction vessel (Appendix A).

Based on the above results, we can conclude that stirring promotes gelatin aggregation in the course of desolvation. Aggregation occurs when low-molecular gelatin fractions are present in sufficient amounts and total gelatin concentration is high (approximately 20 mg/mL and more). These factors make desolvation of medium- and low-bloom gelatins hardly possible to be made in one step when synthesis is carried out in conventional conditions: dropwise addition of poor solvent under stirring. At the same time, the one-time addition of poor solvent with the following short gentle mixing enables the synthesis of gelatin nanoparticles from gelatin with any bloom number.

Dropwise addition of nonsolvent under vigorous stirring is an inevitable part of protein nanoparticle synthesis by the desolvation method. There are many studies reporting a decrease in size and/or polydispersity of albumin [38], silk fibroin [39], α-lactalbumin [40] nanoparticles at higher stirring speeds. The same findings were provided for gelatin nanoparticles by different research groups [41,42]. Conversely, Pei et al. [43] showed that an increase of gelatin concentration in water–ethanol mixture under agitation leads to growth of gelatin nanoparticle size and even to gelation. Removal of low-molecular-weight fractions was not performed in this work as it was aimed at studying gelatin behavior in ethanol–water mixtures rather than the preparation of nanoparticles. Intense shaking provided sedimentation of gelatin 75 bloom in the course of the first desolvation step [44]. There is no contradiction between these reports. Subara and Abdelrady with colleagues removed low-molecular fractions by traditional first desolvation step and studied the effect of stirring speed performing the second desolvation step whereas Pei and colleagues worked with untreated gelatin. Note that Pei et al. [43] used high-bloom gelatins (bloom values of 300 and 320, respectively), which are less susceptible to stirring. We suppose that small amounts of low-molecular fractions in such gelatins can provoke some increase in nanoparticle size, but not aggregation. Additionally, we need to mention a recent paper by Subara and Jaswir in which nanoparticles were synthesized by one-step technique [22] from gelatin hydrolysate comprising small gelatin peptides (about 20 amino acids) [45]. Notably, even at concentrations as high as 20% relatively monodisperse nanoparticles were formed. Therefore, it seems that extremely low gelatin hydrolysate pieces in contrast to low-bloom gelatin allow the preparation of nanoparticles in one step under stirring. 

Stirring-free desolvation was applied for the preparation of monodisperse silk fibroin nanoparticles by Seib et al. [46]. The method is to add aqueous silk fibroin solution in acetone in a drop-by-drop manner. However, the same research group reported decreasing the size of silk fibroin nanoparticles when the addition of protein was performed under stirring [39].

We cannot explain why stirring affects gelatin desolvation. Morel et al., studied a mixing-induced aggregation of wheat gluten and proposed that the formation of disulfide and isopeptide bonds as well as hydrophobic interactions can drive aggregation [47]. There are few cysteine residues in gelatin molecules [4], therefore most likely other mechanisms are involved.

### 3.2. Influence of pH, Gelatin Concentration, Type, and Volume of Desolvating Agent on the Size and Yield of Gelatin Nanoparticles

In earlier works, factors affecting the synthesis of gelatin nanoparticles were extensively studied [20,22,48,49]. However, the authors of these articles used a conventional technique based on the slow addition of desolvating agents to gelatin solution under stirring. We used a modified desolvation method that relies on the one-time addition of non-solvent to the solution of gelatin. Therefore, we decided to re-evaluate how different factors influence the desolvation outcome. Two sets of experiments were carried out. The first part of the experiments was done using very small volumes of gelatin solution (200 μL) and only one type of gelatin. In the second part, five types of gelatin were tested in 20-fold larger volumes. A detailed description of experiments and results can be found in Appendix A. Below we highlight the key findings on the effects of synthesis conditions on nanoparticle properties.

Isopropyl alcohol was a more effective desolvating agent than ethanol and methanol. It provided homogeneous nanoparticle suspensions with considerably higher yields. Isopropyl alcohol has the lowest polarity index and highest dielectric constant in comparison with methanol and ethanol. It has been reported that the desolvation of α-lactalbumin by isopropyl alcohol provided the largest nanoparticles [50]. Moreover, a lower volume of isopropyl alcohol is required to completely desolvate bovine serum albumin in comparison with ethanol [51]. A more complex relationship between properties of non-solvent and its influence on the size of nanoparticles, not limited to the difference of dielectric constants, was demonstrated by Mohammad-Beigi et al. [52]. Pei et al., assumed that differences in alcohols’ viscosity can influence the desolvation of gelatin [43]. As a rule, smaller nanoparticles were obtained using ethanol, which is in line with the previous reports [51]. This difference, though, was more distinct at the gelatin concentration of 18 mg/mL. At lower gelatin concentrations, especially at higher pH, ethanol provided a very low degree of gelatin to nanoparticle transformation, which led to unstable DLS results. Larger volumes of alcohols resulted in higher yields due to the lower solubility of gelatin at higher alcohol concentration. Diameters of nanoparticles became lower, probably because the addition of large alcohol volume decreased the final gelatin concentration. Similar trends were observed by Shamarekh et al. [26].

An increase in gelatin concentrations led to the growth of nanoparticle size. This effect was more prominent for gelatins B and fish gelatin at pH 11. The same relationship was reported by different researchers [22,23,24,26,43]. Probably, a higher local concentration of gelatin favors the desolvation process, which is also illustrated by higher particle yields at the higher protein concentration [26]. However, other factors, such as gelatin solution viscosity can also impact desolvation results [43].

Higher yields were observed for the same ethanol volume for gelatin B with bloom values of 225 in comparison with gelatin B bloom 75 which is in line with the results obtained by Nixon et al. [53], who showed that lower ethanol volume is necessary to initiate coacervation of gelatins with higher bloom numbers. Interestingly, an opposite relationship was observed for isopropyl alcohol.

Electrostatic repulsion of gelatin molecules and pH-dependent degree of molecule hydration influence both size and yield of nanoparticles as was shown by many researchers [22,29,54,55]. Generally, higher pH values (far away from gelatin isoelectric point) resulted in lower yields and smaller particles.

Obtained results demonstrate that control over synthesis parameters enables tuning of nanoparticles’ properties and process yield.

### 3.3. Synthesis of Nanoparticles from Different Gelatins in Hundreds-of-Milligram-Scale

In order to demonstrate the scope of the modified desolvation method, we synthesized nanoparticles from various types of gelatin with different bloom numbers. On the basis of optimization experiments, we decided to desolvate gelatins with isopropyl alcohol to obtain high yields of nanoparticles. We intended to prepare nanoparticles with hydrodynamic diameters less than 200 nm, hence 10 mg/mL gelatin solutions with high pH were used. 

Scalability is an essential part of nanoparticle products implementation. Ideally, the synthesis procedure should be not only scalable but also reproducible, providing small batch-to-batch variability [56]. Optimization experiments were done using rather small portions of gelatin (less than 80 mg). Here we performed the hundreds-of-milligram-scale synthesis of gelatin nanoparticles by modified desolation method. 

We synthesized nanoparticles from porcine, bovine, and fish gelatin with different bloom numbers including the lowest available (62 and 75). The total initial amount of gelatin was 1000 mg, three batches were synthesized for each kind of gelatin. Each batch had an ID, indicating the type of gelatin source and number of replication, e.g., “B225-2”. Key steps of the synthesis procedure are presented in Figure 3. Isopropyl alcohol was added to gelatin solution, the resulting suspension of gelatin nanoparticles was kept in the water bath for 30 min, then nanoparticles were stabilized by glutaraldehyde, washed by centrifugation, concentrated, and sonicated. Properties of synthesized nanoparticle batches are summarized in Table 1 and Figure 4. In total, we confirmed that the modified desolvation method enables nanoparticle synthesis from different gelatin types with various bloom numbers.

#### 3.3.1. Size, Zeta Potential, and Shape of Nanoparticles

The hydrodynamic diameter of most nanoparticles was between 130 and 160 nm. The lowest nanoparticles were prepared from gelatin B, 225 bloom. We assessed the reproducibility of nanoparticle synthesis by calculating coefficients of variation (CV) for each type of gelatin and comparing it with available literature data. Reproducibility was the lowest for nanoparticles prepared from gelatin B, 75 bloom (CV = 7.2%), whereas for other gelatin types CVs were from 1.2% to 4.4%. Reproducibility of the preparation of recombinant human serum albumin nanoparticles by desolvation method was studied by Langer et al. [57]. Three batches were prepared; CV was 9.1% [57]. Gelatin nanoparticles of different sizes prepared by optimized two-step desolvation were reported by Dr. Claus Zwiorek [20]. Six batches were synthesized for each type of gelatin nanoparticles; coefficients of variation were 3.4% (mean size is 300 nm), 1.4% (150 nm), and 13.2% (100 nm). Thus, the modified desolvation method enables the reproducible synthesis of gelatin nanoparticles. Polydispersity indices were lower than 0.2 for all batches and lower than 0.1 for most batches, indicating that synthesized nanoparticles had homogeneous size distribution.

The zeta potential of nanoparticles was measured in a neutral phosphate buffer, pH 7. According to information from the manufacturer, the isoelectric point is 4.7–5.3 for gelatins B, 7.0–9.5 for gelatins A, and 6.0 for fish gelatin. Nanoparticles prepared from gelatin B had the lowest zeta potential of about −11 mV, whereas nanoparticles made from fish gelatin and gelatin A had more positive zeta potential: from −7 to −9 mV. Notably, the zeta potential of gelatin nanoparticles is much lower than the conditional stability threshold of ±30 mV [58], indicating that forces other than electrostatic repulsion provide their colloidal stability, which is confirmed by the results of their detailed colloidal stability study (see Section 3.4).

Scanning electron microscopy and transmission electron microscopy (TEM) showed that nanoparticles prepared from all types of gelatin had a round shape (Figure 5). The insufficient quality of SEM photographs did not allow us to measure their sizes. Nevertheless, a visual assessment of the photos demonstrated that the sizes of most of the particles are in the range of 100–200 nm, which coincides with the DLS results. The mean diameter of B75-1 obtained by TEM was lower (100 ± 21 nm) than the hydrodynamic diameter (Appendix A), which is explained by the well-known tendency of DLS to overestimate the size of nanoparticles. Microscopy demonstrated the presence of large nanoparticles (they can be seen in Figure 5B) and some amount of aggregates, however, in general nanoparticles were homogeneous.

#### 3.3.2. Yield

We obtained stable aqueous suspensions of gelatin nanoparticles having volumes of 44–46 mL and containing from 13.8 to 18.2 milligrams of nanoparticles per milliliter. Therefore, the method allows the preparation of 600–800 mg of nanoparticles in 6–7 h. This value can be increased by the application of larger reagent volumes or by changing the synthesis conditions: decreasing the pH, increasing the gelatin concentration, and so on (see Appendix A). The yield of synthesis (degree of gelatin-to-nanoparticles conversion) was between 62 and 82%, which is higher than reported for two-step and one-step desolvation: 1.5–62% (Appendix A). However, in special conditions yields of conventional one- and two-step desolvation procedures can reach 70–80% [22,48,59]. For the nanoprecipitation method yields as high as 90 ± 5% were reported [60], however, lower yields were obtained in other works (Appendix A). Based on optimization experiments, we claim that yields up to 95% can be reached with the aid of the modified desolvation method due to the high desolvating efficiency of isopropyl alcohol. 

In the desolvation technique, there is sometimes a trade-off between the yield and size of nanoparticles. Conditions that favor protein desolvation provide better yields, but the larger size of nanoparticles. An increase in yield can be achieved by lowering the pH or by an increase of added alcohol volume. The second approach enables higher yields and even lower sizes, but at the expense of reaction volume increase. In this work, we synthesized gelatin nanoparticles at high pH (10 for gelatin B and 11 for fish gelatin and gelatin A), besides non-solvent to gelatin volume ratio was 5. Using lower pH and/or larger ratios, higher yields could be achieved.

One more thing that needs to be explained is lower yields obtained for B225 batches. We think that losses of nanoparticles during washing steps can be the reason. Nanoparticles B225-1/2/3 had the lowest diameters and required more time to complete sedimentation. When decantation of the supernatant was performed, the loose part of the sediment was removed. This was observed for all batches but in the case of B225-1/2/3 it was the most pronounced.

#### 3.3.3. Absorbance and Fluorescence Spectra of Gelatin Nanoparticles

Protein nanoparticles cross-linked with glutaraldehyde emit fluorescence when excited by UV or visible light [30,61]. Autofluorescence of gelatin nanoparticles can be explained by the presence of C=C (resulting from glutaraldehyde polymerization) and C=N bonds (in the Shiff bases) in nanoparticles’ structure [62]. Fluorescent properties of gelatin nanoparticles can be utilized in bioimaging and biosensing [61]. Moreover, intrinsic fluorescence of nanomaterials could be used to measure their cellular uptake [63,64] and underlines nanoparticle interference with various fluorescent techniques (an example of such interference can be found in Section 3.8). Given that gelatin nanoparticles do not have distinct absorbance peaks (Appendix A), we recorded the fluorescence spectra of nanoparticles at excitation wavelengths from 260 to 560 nm. Nanoparticles prepared from all gelatins possess broad fluorescent peaks, which are red-shifted with the increase of excitation wavelength (Appendix A). Further, we used fluorescent properties of gelatin nanoparticles to assess the change of their structure after the sterilization procedure (see Section 3.7).

The color of gelatin nanoparticles depended on synthesis conditions. Desolvation by ethanol at any pH values or by isopropyl alcohol at pH less than 11 resulted in yellowish suspension. Nanoparticles prepared at pH 11 using isopropyl alcohol were reddish, indicating possible differences in the chemical structure.

### 3.4. Stability of Nanoparticles at Various pH and Salt Concentrations

Colloidal stability of nanoparticles at various pH and in solutions with high ionic strength is highly desirable for practical applications. Conjugation of nanoparticles with different molecules, including recognition molecules (e.g., monoclonal antibodies) as well as a surface modification with stealth or protecting polymers, are usually performed at pH and salt concentration, which are optimal for a specific technique. Therefore, we tested the colloidal stability of nanoparticles in buffers with pH ranging from 4 to 10 measuring their size by DLS immediately after addition to the buffer, then at days 1 and 7. Gelatin nanoparticles prepared from gelatin B, fish gelatin, and gelatin A 180 bloom were stable for one week in all buffers (Figure 6). A slight increase of size and polydispersity was, though, observed in several samples. In these samples usually, one of three technical replicates indicated the presence of aggregates, whereas other replicates showed homogeneous size distribution. Most likely, insignificant aggregation took place; however, most of the particles were in the non-aggregated state. These results coincide with literature data, indicating that the gelatin shell provides good colloidal stability in the wide range of pH values [65].

Nanoparticles prepared from gelatin A with a bloom value of 62 were the only type of nanoparticles for which pronounced aggregation was observed (Figure 6D). These nanoparticles quickly aggregated being exposed to pH 4, but not in other buffers. The relationship between zeta potential and pH was different for nanoparticles prepared from various types of gelatin (Appendix A). Nanoparticles B75 and B225 had more negative zeta potential at a pH range from 5 to 9 which is explained by the difference in isoelectric points between gelatins [49]. 

High salt concentrations had no significant effect on the size of gelatin nanoparticles. It was previously shown that low (7–50 mM), but not high (300 mM) NaCl concentrations promote aggregation of gelatin nanoparticles [66]. We did not observe any signs of nanoparticle aggregation in presence of salts (Appendix A).

We should note that the concentration of nanoparticles was as low as 50 μg/mL. Perhaps, more pronounced aggregation could be detected at higher nanoparticle concentrations. For nanoparticles A180-1/2/3 we detected aggregation at a low nanoparticle concentration (lower than 80 μg/mL) in water, but not in phosphate buffer. Glycine treatment stabilized these nanoparticles even at low concentrations. We cannot explain this phenomenon, moreover, it was not observed for nanoparticles prepared from other types of gelatin.

### 3.5. Storage Stability

The stability of nanoparticles upon storage is necessary for their practical application in any field. For end users concentrated aqueous suspension is, perhaps, the most convenient form of nanoparticle preparations. We studied the size and structural integrity of gelatin nanoparticles prepared using a modified desolvation method after 4 weeks of storage in water at +4 °C. Surprisingly, a decrease in hydrodynamic diameter and polydispersity indices were detected for all batches (Figure 7). Moreover, after 4 weeks of storage batch-to-batch variability of nanoparticle sizes also became lower. These unexpected results contradict the data reported in the literature. Previous works reported no change or growth of nanoparticle diameter [19,67]. The decrease of hydrodynamic diameter of gelatin nanoparticles stored in the lyophilized state was observed and explained by incomplete rehydration [68], however, in our study nanoparticles were stored in water and did not change hydration state. We supposed that partial dissolution of nanoparticles could occur. The concentration of free protein in nanoparticle suspension, as well as turbidity of nanoparticle suspension, were monitored (Appendix A). Turbidity can reflect both the change of nanoparticle size and their dissolution. For almost all nanoparticles, a decrease in suspension turbidity was observed. The concentration of free gelatin was also higher on the 28th day. However, the relative amount of free protein did not exceed 1% of total gelatin.

It can be assumed that the storage of gelatin nanoparticles is accompanied by partial disintegration. Between-batch size variability after 4 weeks of storage became lower, indicating that this process affects larger nanoparticles and aggregates. The percentage of free gelatin was quantified by centrifugation of nanoparticles at 20,000 *g* and measurement of protein in supernatant. Mentioned speed is not high enough to pellet small nanoparticles (say, 10–20 nm). Therefore, the slight increase of protein concentration in supernatants can be explained by both the release of single gelatin molecules and the decomposition of larger nanoparticles into smaller ones. The presence of a certain amount of smaller nanoparticles (in contrast to larger nanoparticles) cannot affect the results of DLS measurements, because their light scattering ability is too small. Taking into account that the degree of turbidity decrease did not correlate with the degree of the particle diameter decrease, we assume that partial decomposition of nanoparticles into smaller nanoparticles took place.

Literature data suggest that glutaraldehyde cross-linking produces stable gelatin nanoparticles [19,67], however, in most cases, only particle size but not other properties were assessed. We examined the storage stability of non-sterile nanoparticle preparations, which were stored in deionized water without any preliminary physical or chemical treatment. Therefore, bacterial contamination and protease activity could play a role in nanoparticle degradation. Undoubtedly, further study of nanoparticle stability needs to be conducted. However, even if nanoparticles are unstable in suspension, there are optimized methods of their storage in freeze-dried conditions [68].

### 3.6. Loading of Gelatin Nanoparticles with Fluorescent Complex

Biomedical and biotechnological applications of gelatin nanoparticles require them to be loaded with a wide spectrum of therapeutic and imaging agents: small molecules, nanomaterials, and polymers. We tested whether the modified desolvation method is appropriate for incorporation of model hydrophobic substance: fluorescent complex containing europium ion and two chelating ligands, namely 1,10-phenanthroline and 4-(4-Methylphenyl)-2,4-dioxobutanoic acid. Europium and ligands were dissolved in ethanol; then gelatins of each type were desolvated with the resulting solution. Ethanol was chosen because of the insufficient solubility of fluorescent complexes in isopropyl alcohol. Europium complexes possess bright fluorescence facilitating confirmation of successful loading, besides, they are insoluble in water resembling small molecules used in drug delivery. Moreover, the long-living and large-Stokes-shift fluorescence of europium complexes are easily distinguishable from the inherent fluorescence of gelatin nanoparticles.

Nanoparticles synthesized from all the gelatin types were loaded with fluorescent complexes (Figure 8). A narrow peak (600–630 nm) of europium emission was detected in suspensions of purified gelatin nanoparticles after desolvation with an ethanol solution containing fluorescent complexes (Appendix A) but was not detected in bare gelatin nanoparticles (Figure 8). Supernatants obtained during the purification of fluorescent gelatin nanoparticles displayed weak fluorescence, which, nevertheless, was negligible in comparison with that of nanoparticles. Therefore, europium complexes were associated with gelatin nanoparticles, however, we did not study whether they were located on the surface or were embedded in the nanoparticle body. Weak fluorescence in supernatants was most likely due to the leakage of fluorescent complexes induced by ultrasound treatment which accompanied nanoparticle purification.

We should note that sub-micro particles and microparticles rather than nanoparticles were synthesized from fish gelatin and type A gelatins (Figure 9A). Type B gelatin 75 bloom yielded the highest quality nanoparticles with the lowest polydispersity indices (Figure 9). The mean diameter of these nanoparticles was 115 ± 23 nm (Appendix A), being slightly larger than unloaded ones.

In this work, we did not optimize loading conditions, only a couple of preliminary experiments were carried out with type B gelatins. We suppose that proper optimization can enable the preparation of 100–200 nm sized nanoparticles loaded with hydrophobic molecules from any type of gelatin.

### 3.7. Sterilization of Gelatin Nanoparticles

Microbial contamination of gelatin nanoparticles is undesirable for almost all applications. The presence of microorganisms or their fragments in the nanoparticle preparations could pose a risk for patients, besides, microbial enzymes can destroy nanoparticles and decrease their shelf-life [69]. Geh et al., reported that autoclave sterilization of gelatin nanoparticles leads to the partial release of gelatin molecules and slight nanoparticle growth, whereas the higher cross-linking degrees and milder autoclaving conditions make nanoparticles less sensitive to thermal degradation [68]. At the same time, Ma et al., successfully autoclaved bovine serum albumin nanoparticles prepared by the desolvation method [70]. Based on these results we decided to sterilize gelatin nanoparticles by autoclaving in the mildest conditions: 15 min at 0.5 atm above atmospheric pressure. Control (non-autoclaved) nanoparticles were kept at +4 °C.

In autoclaving experiments, we used gelatin nanoparticles treated with glycine. Although we added an excess of glutaraldehyde in relation to the number of primary amines, a small portion of unreacted amino groups can remain on the outer surface of nanoparticles [59]. Glycine quenched free surface carbonyl groups and supposedly decreased the probability of their reaction with remaining primary amines. Unfortunately, glycine treatment led to unstable DLS measurements. DLS results were different even between technical replicates: for some samples, one or two of three measurements indicated the presence of aggregates or increase of average diameter, whereas other measurements showed narrow distribution. This feature of the DLS method was previously well illustrated by Langevin and co-authors [71]. They put TiO_2_ nanoparticles in a buffer with high ionic strength and measured the diameter of nanoparticles by DLS and differential centrifugal sedimentation, which is much less sensitive to the presence of aggregates than DLS. A large number of aggregates was detected by DLS, on the contrary, differential centrifugal sedimentation showed that the agglomeration degree was not significant. 

To obtain more stable DLS results we centrifuged autoclaved nanoparticles for 10 min at 1000 *g* before the size measurements (Figure 10). We suppose that slight aggregation occurred, however, most of the particles withstood sterilization and kept their original size because no visible signs of aggregation were detected (Appendix A). The presence of aggregates and free gelatin may be incompatible with applications, which require very homogeneous nanoparticle preparations (e.g., drug delivery). However, autoclaving may be a good and simple choice in the fields with less strict requirements to size distribution. Aggregates can be quickly removed by low-speed centrifugation using commonly available sterile centrifuge tubes. 

The influence of autoclave sterilization on the integrity of gelatin nanoparticles was studied by two methods. 

Firstly, we measured the concentration of free protein in nanoparticle suspensions before and after autoclaving. Nanoparticles were pelleted by centrifugation, and supernatants were analyzed by bicinchoninic acid assay. Generally, after autoclave sterilization, the percentage of free gelatin molecules did not exceed 1–2% of total nanoparticle weight, which is, though, 2–5-fold higher in comparison with untreated nanoparticles (Figure 11A). Partial degradation and aggregation of gelatin nanoparticles upon autoclaving are in line with previous reports [68]. 

Secondly, we assessed the turbidity of nanoparticle preparations. Recently, Geh with colleagues observed a pronounced decrease of turbidity after sterilization of gelatin nanoparticles in the autoclave [68]. For almost all batches of nanoparticles, slight (not more than 8%) decrease in turbidity was observed. Both changes in size and dissolution could affect turbidity (Figure 11C). Taking into account DLS data and free protein change, we suppose that dissolution of a small percentage of nanoparticles can take place during sterilization.

Another issue is a chemical alteration caused by sterilization. Autoclaving leads to a decrease of cross-linking degree and an increase of free amino group content in gelatin nanoparticles [68]. Indeed, the zeta potential of sterilized nanoparticles was a bit more positive (by 1–2 mV) compared to plain nanoparticles (Figure 11B). Moreover, the color of nanoparticle suspensions changed after sterilization (Appendix A). The formation of new chemical bonds in gelatin nanoparticles was indirectly assessed by studying their fluorescence behavior at different excitation wavelengths. We compared emission spectra of gelatin nanoparticles before and after sterilization. Excitation wavelengths varied from 260 to 560 nm. Emission spectra recorded at excitation wavelengths of 300, 400, and 500 nm are presented in Appendix A. Excitation at 300 and 400 nm resulted in a strong increase of emission, whereas excitation at 500 nm had almost no effect. These results confirm that some structural alterations accompanied autoclaving. Change of emission was not caused by evaporation of nanoparticle suspension, otherwise, an increase of emission should also be observed after excitation at 500 nm. Moreover, the volume of nanoparticle suspension was controlled in the course of the experiment.

Although autoclaved samples retained their key properties, we should note that autoclaving of gelatin nanoparticles is appropriate mainly for food applications or in other fields which do not require parenteral usage of nanoparticles. Autoclaving kills microorganisms but does not remove endotoxin, which can interfere with cell and tissue culturing and provokes immune system activation or even endotoxin shock in animals [69]. Autoclaving may be useful for applications where the presence of a small percentage of aggregates and free protein is acceptable. For pharmaceutical purposes, the whole synthesis can be performed in aseptic conditions with endotoxin-free reagents. Another option is post-synthesis depyrogenation and sterilization by gamma-irradiation [72], which was shown to be compatible with gelatin nanoparticle formulations.

### 3.8. Effect of Gelatin Nanoparticles on the Viability of Peripheral Blood Mononuclear Cells

Application of gelatin nanoparticles in drug/gene/antigen delivery or bioimaging requires their blood circulation, making interaction with blood components unavoidable. Therefore, the toxicity of gelatin nanoparticles towards blood immune cells was studied. Peripheral blood mononuclear cells (PBMC) are a complex mixture of cells with a round-shaped nucleus comprising regulatory and effector cells, namely T cells, B cells, NK cells, and monocytes. Being primary cells, they represent the reaction of human blood cells to nanoparticles in a more natural way in comparison with available lymphocyte cell lines [73]. PBMC were isolated from the blood of three healthy donors and incubated with glycine-quenched sterilized gelatin nanoparticles for 24 h. Dead cells were stained with propidium iodide and quantified by flow cytometry. The range of gelatin nanoparticles concentrations was based on literature data, indicating that the concentration of nanoparticles in the blood can reach values of hundreds of micrograms per milliliter after parenteral administration [74].

The viability of PBMC was higher than 88% in a whole range of nanoparticle concentrations, indicating low cytotoxicity of gelatin nanoparticles (Figure 12). However, we should note that nanoparticles engulfed or absorbed by cells interfere with fluorescence measurements. Nanoparticles emit fluorescence between 543 and 627 nm (Appendix A) (propidium iodide detection channel), moreover, at higher nanoparticle concentrations PBMC uptake (or absorb) more nanoparticles as can be seen by the increase of cell granularity (which was assessed by measuring the side scattering of cells) (Appendix A). Therefore, a more thorough study of gelatin nanoparticle toxicity is to be made in the future.

### 3.9. Future Perspectives and Limitations of This Study

The modified desolvation method makes it possible to synthesize nanoparticles from gelatin regardless of its origin and bloom number. This method can be a facile alternative to the commonly utilized two-step desolvation method. Noteworthy, in the course of nanoparticle synthesis, we used common laboratory equipment: centrifuge with a capacity less than 500 mL, thermostat, water bath, and sonicator. All the reagents used are readily available and cheap. Synthesis procedures are simple and can be performed even by non-trained personnel. 

One can conclude that the proposed method works well in a laboratory, but is it suitable for large-scale nanoparticle manufacturing? To meet the needs of market consumers, nanoparticles manufacturing rates should be as high as kilograms per day [75,76]. Theoretically, this goal can be achieved by a straightforward hundredfold increase of synthesis volumes and by performing multiple parallel syntheses. Undoubtedly, volume increase can lead to a change of nanoparticles size, yield, and their properties caused by mass transfer rate alterations [76]. However, we emphasized that at certain gelatin concentrations mixing rate has a very small effect on nanoparticles size. Moreover, as one can see, gelatin does not aggregate being quickly mixed with a large volume of non-solvent (in contrast to, say, bovine serum albumin). In our opinion, these factors favor the scaling-up of the proposed modified desolvation method.

Recently, direct addition of non-solvent to gelatin under gentle mixing has been realized in the microfluidics-assisted method [77]. This approach allows continuous manufacturing of gelatin nanoparticles. Microfluidic devices enable large-scale synthesis of nanoparticles and better control over their preparation [78,79], however, scaling-up can be challenging as it requires chip parallelization or channel diameter increase [39]. The high costs of setting up the microfluidic devices for industrial scale-up, as well as technical issues (e.g., channels clogging [78,80]), are among potential barriers to their implementation. Thus, despite numerous advantages of microfluidics technology, batch methods (including modified desolvation method) are still attractive in terms of their industrial application [39].

In this study, glutaraldehyde was utilized for nanoparticle cross-linking. Despite being effective and widely available, it raises concerns about its potential toxicity (both as a component of nanoparticles and a component of waste). Therefore, the toxicity of gelatin nanoparticles stabilized with glutaraldehyde needs to be thoroughly tested. Moreover, other approaches to gelatin cross-linking should be considered, e.g., reagentless irradiative cross-linking [81] or stabilization by polymer entrapment [3]. 

The environmental impact of all nanoparticle synthesis components should also be considered before their translation into practice [82]. The main concerns are usually related to the toxicity of cross-linking agents, however, proper management of organic waste comprising organic solvent itself and products of its interaction with the cross-linking agent can also be a problem. More cheap and effective (from the synthesis point of view) solvent can be less appropriate in terms of disposal, therefore, different combinations of desolvating agents (not necessarily organic solvents!) and cross-linking agents are to be tested. Here, we used short-chain alcohols as desolvating agents. They can be classified as environmentally favorable solvents [83]. However, other organic solvents or other approaches to desolvation (e.g., salting out), which can be potentially more effective or safe, need to be studied in the future. Moreover, testing of different cross-linking agents or non-solvents is of importance, because some substances to be loaded in gelatin nanoparticles can be incompatible with specific synthesis conditions, i.e., non-soluble in particular desolvating agent, unstable at high pH, and so on.

In conclusion, we need to mention several limitations of the present study:

The size of all synthesized gelatin nanoparticles exceeded 100 nm. We did not obtain smaller nanoparticles, however, the desolvation method allows the preparation of nanoparticles whose diameter is less than 100 nm [23,26]. We suppose that a decrease of initial gelatin concentration or increase of gelatin solution pH is a possible way to obtain nanoparticles smaller than 100 nm.We did not prepare nanoparticles from gelatin solutions with concentrations higher than 20 mg/mL at a high scale. As we mentioned in Section 3.3, our goal was to prepare relatively small nanoparticles, less than 200 nm, which is possible by using smaller gelatin concentrations for all tested gelatin types. Data obtained in the course of optimization experiments and our previous results [30] both demonstrate that gelatin nanoparticles can be prepared at high starting gelatin concentrations. Variation of pH and volume of the desolvating agent is a possible way to decrease the size of nanoparticles when the concentration of gelatin is high.As we mentioned before, we did not remove endotoxin from gelatin nanoparticles nor examine endotoxin concentration in nanoparticle preparations prior to cell viability testing. Synthesis of apyrogenic nanoparticles is a challenging task, which requires a separate set of experiments and was, therefore, beyond the scope of the present work. We just note that the protocol of gelatin depyrogenation was previously reported by Singh et al. [37], besides, post-synthesis depyrogenation by gamma-irradiation also remains a possible option [72].The effect of several factors on the desolvation process was not studied: temperature of starting materials [49], salt concentration, acidic pH values, gelatin pre-incubation [84], the longevity of incubation with alcohols, and so on. Nevertheless, we performed preliminary experiments adding NaCl before desolvation. The addition of salt resulted in the formation of microparticles visible by the eye, however, a systematic study has not been conducted.Despite various types of animal gelatin being tested, we did not prepare nanoparticles from human recombinant gelatin. Application of natural gelatin from animal sources can be limited due to pathogen (first of all, prions) contamination, religious reasons, and its potential immunogenicity [4]. In previous papers, gelatin nanoparticles synthesized from recombinant human gelatin by one-step desolvation method were described [23], therefore, we believe that the universal nature of the proposed method enables usage of human recombinant gelatin as a starting material.

## 4. Conclusions

In this work, a simple and scalable modified desolvation method to fabricate gelatin nanoparticles was proposed. The method allows the preparation of gelatin nanoparticles with sizes 100–300 nm with a high yield on the hundreds-of-milligram scale. The resulting nanoparticles possess excellent colloidal stability, low cytotoxicity, and can be loaded with hydrophobic molecules, being appropriate for application in food science, drug and vaccine delivery, cell culture, and immunoassays development.

## Figures and Tables

**Figure 1 pharmaceutics-13-01537-f001:**
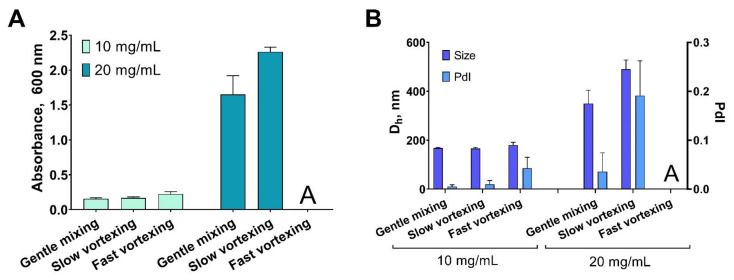
Influence of stirring intensity of the turbidity (**A**) and size (**B**) of gelatin nanoparticles. Dh—hydrodynamic diameter; PdI—polydispersity index. Mean values of three individual batches are shown, mean ± SD. Symbol “A” stands for aggregation.

**Figure 2 pharmaceutics-13-01537-f002:**
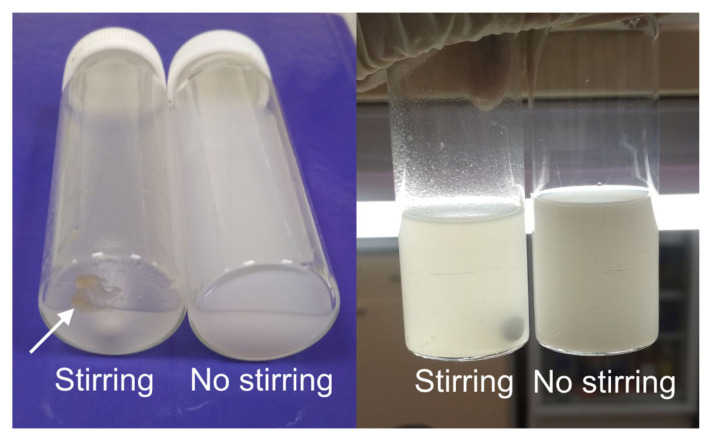
Gelatin nanoparticles prepared from gelatin B, 75 bloom by one-time addition of ethanol without stirring and by dropwise addition of ethanol under stirring. Large gelatin aggregates are labeled with arrows.

**Figure 3 pharmaceutics-13-01537-f003:**
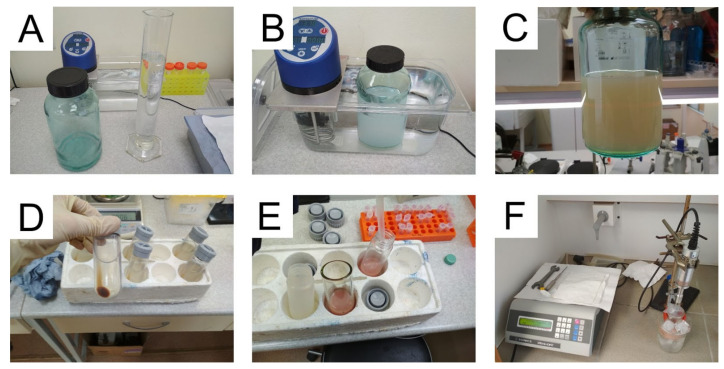
Stages of gelatin nanoparticle synthesis. (**A**) Solution of gelatin and isopropyl alcohol before mixing; (**B**) incubation of nanoparticles at the water bath; (**C**) nanoparticle suspension appearance after glutaraldehyde addition; (**D**) sediment of nanoparticles after centrifugation; (**E**) combining of washed gelatin nanoparticles prior to final sonication; (**F**) sonication of gelatin nanoparticles on ice.

**Figure 4 pharmaceutics-13-01537-f004:**
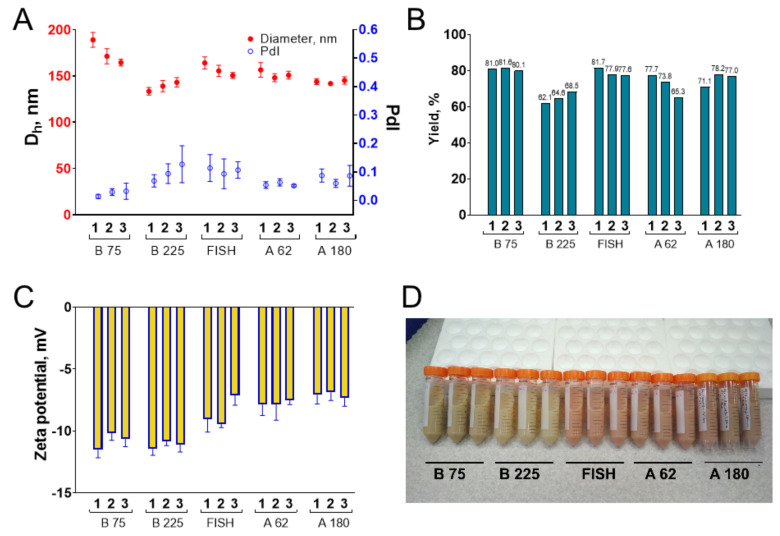
Properties of gelatin nanoparticles prepared from various gelatins in the hundreds-of-milligram scale: (**A**) size and polydispersity index; (**B**) yield; (**C**) zeta potential (at a pH 7); (**D**) the appearance of gelatin nanoparticle suspensions. Numbers 1, 2, and 3 denote the batch numbers. Dh—hydrodynamic diameter; PdI—polydispersity index. Mean values of three technical replicates are shown, mean ± SD.

**Figure 5 pharmaceutics-13-01537-f005:**
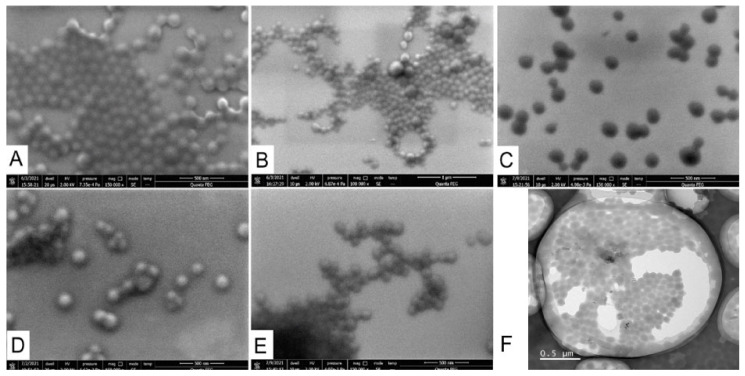
SEM (**A**–**E**) and TEM (**F**) images of gelatin nanoparticles. (**A**,**F**) B75-1; (**B**) B225-1; (**C**) FISH-1; (**D**) A62-1; (**E**) A180-1. Scale bars are 500 nm (**A**,**C**–**F**) or 1000 nm (**B**).

**Figure 6 pharmaceutics-13-01537-f006:**
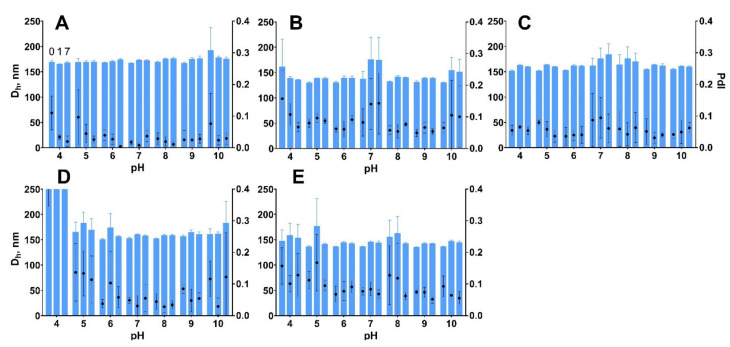
Colloidal stability of gelatin nanoparticles at different pH values. (**A**) B75-1; (**B**) B225-1; (**C**) FISH-1; (**D**) A62-1; (**E**) A180-1. Dh—hydrodynamic diameter; PdI—polydispersity index. Mean values of three technical replicates are shown, mean ± SD. Measurements were performed on days 0, 1, and 7.

**Figure 7 pharmaceutics-13-01537-f007:**
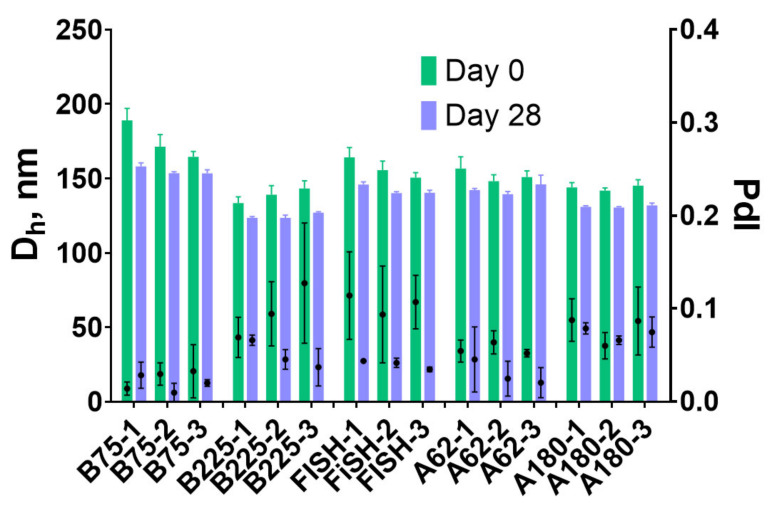
Size (Dh) and polydispersity index (PdI) of gelatin nanoparticles after the synthesis and in 28 days. Mean values of three technical replicates are shown, mean ± SD.

**Figure 8 pharmaceutics-13-01537-f008:**
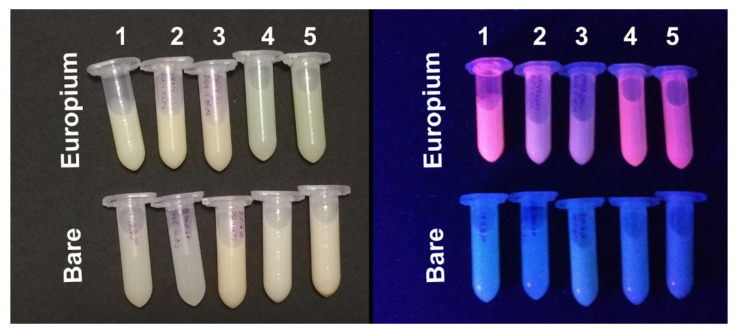
Gelatin nanoparticles containing and not containing europium complexes at daylight (**left**) and in UV light (360 nm, **right**). Nanoparticles were prepared from gelatin: 1—B, 75 bloom; 2—B, 225 bloom; 3—FISH; 4—A, 62 bloom; 5—A, 180 bloom.

**Figure 9 pharmaceutics-13-01537-f009:**
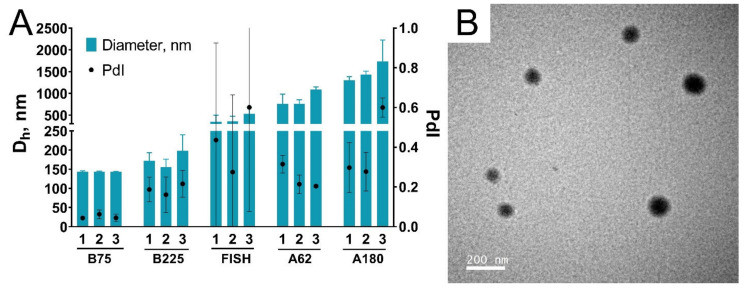
Size of gelatin nanoparticles loaded with fluorescent europium complexes (**A**) and TEM image of europium-loaded nanoparticles prepared from gelatin B 75 bloom (**B**). Dh—hydrodynamic diameter; PdI—polydispersity index. Mean values of three technical replicates are shown, mean ± SD. The scale bar is 200 nm.

**Figure 10 pharmaceutics-13-01537-f010:**
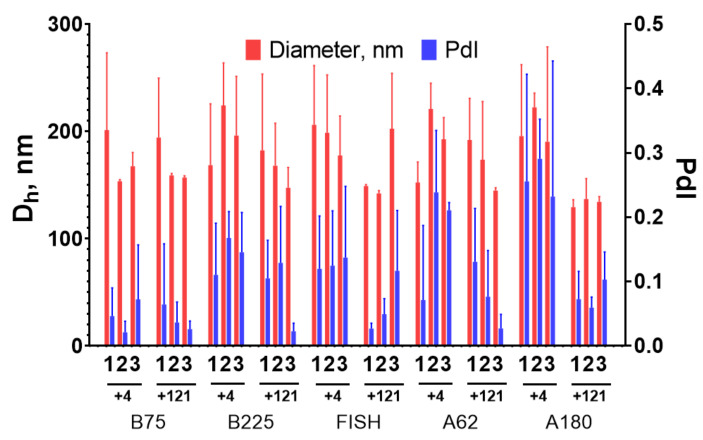
Size (Dh) and polydispersity index (PdI) of non-autoclaved (+4) and autoclaved (+121) gelatin nanoparticles. Mean values of three technical replicates are shown, mean ± SD.

**Figure 11 pharmaceutics-13-01537-f011:**
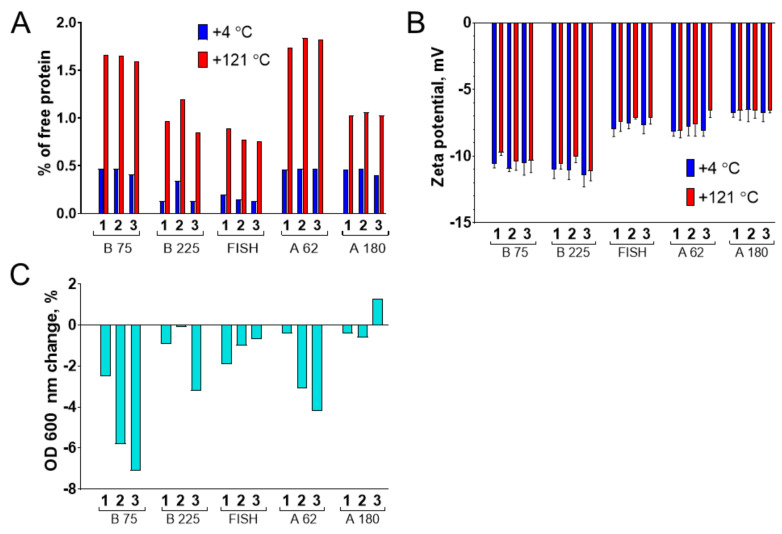
Properties of sterilized (+121 °C) and non-sterilized (+4 °C) nanoparticles. (**A**) The percentage of free protein in relation to nanoparticle concentration; (**B**) zeta potential at a pH 7; (**C**) change of turbidity of the gelatin nanoparticle suspensions after sterilization. Mean zeta potential values of three technical replicates are shown, mean ± SD.

**Figure 12 pharmaceutics-13-01537-f012:**
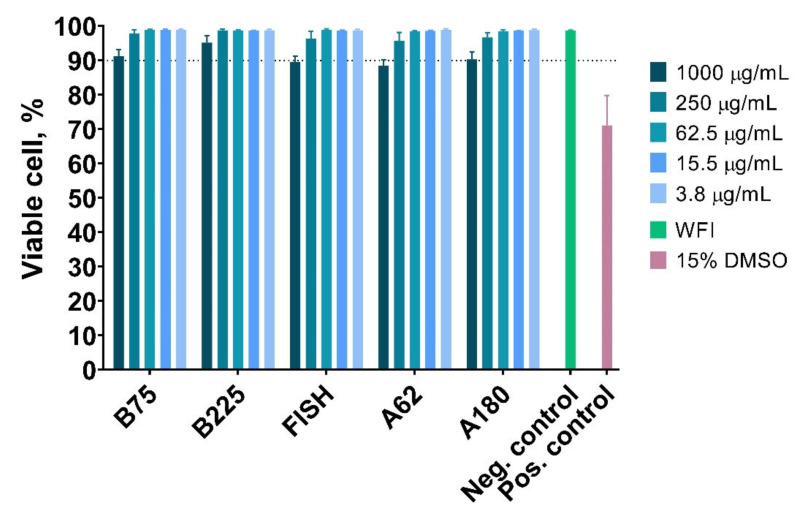
Viability of PBMC in the presence of various concentrations of gelatin nanoparticles. WFI—water for injection. PBMC from three volunteers were obtained, mean ± SD.

**Table 1 pharmaceutics-13-01537-t001:** Properties of gelatin nanoparticles prepared from different gelatins.

Batch ID	Suspension Volume, mL	Concentration, mg/mL	Total Dry Weight of Nanoparticles, mg ^1^	Yield, %	D_h_, nm ^1^	PdI ^2^	Zeta Potential, mV
B75-1	45	18.0	810.0	81.0	189 ± 8 ^3^	0.014 ± 0.007	−11.5 ± 0.6
B75-2	46	17.7	815.6	81.6	171 ± 8	0.030 ± 0.012	−10.2 ± 0.6
B75-3	44	18.2	800.8	80.1	165 ± 4	0.033 ± 0.029	−10.7 ± 0.6
B225-1	45	13.8	621.0	62.1	133 ± 4	0.069 ± 0.022	−11.5 ± 0.5
B225-2	45	14.4	646.2	64.6	139 ± 6	0.094 ± 0.034	−10.9 ± 0.3
B225-3	45	15.2	685.4	68.5	143 ± 5	0.127 ± 0.065	−11.1 ± 0.6
FISH-1	46	17.8	817.0	81.7	164 ± 7	0.114 ± 0.047	−9.0 ± 1.0
FISH-2	46	16.9	778.8	77.9	156 ± 6	0.094 ± 0.052	−9.4 ± 0.3
FISH-3	46	16.9	775.6	77.6	151 ± 3	0.107 ± 0.029	−7.1 ± 0.8
A62-1	46	16.9	777.4	77.7	157 ± 8	0.054 ± 0.012	−7.9 ± 0.9
A62-2	45	16.4	738.0	73.8	148 ± 4	0.064 ± 0.012	−7.8 ± 1.3
A62-3	45	14.5	652.5	65.3	151 ± 4	0.052 ± 0.004	−7.5 ± 0.4
A180-1	45	15.8	711.0	71.1	144 ± 3	0.088 ± 0.023	−7.1 ± 0.7
A180-2	46	17.0	782.0	78.2	142 ± 2	0.060 ± 0.014	−6.9 ± 0.7
A180-3	45	17.1	769.5	77.0	145 ± 4	0.087 ± 0.036	−7.4 ± 0.6

^1^ Hydrodynamic diameter. ^2^ Polydispersity index. ^3^ Mean of three technical replicates ± standard deviation.

## Data Availability

The datasets used and/or analyzed during the current study are available from the corresponding author on reasonable request.

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
