# Peer review of "Modified Desolvation Method Enables Simple One-Step Synthesis of Gelatin Nanoparticles from Different Gelatin Types with Any Bloom Values"

_pharmaceutics, 2021, doi:10.3390/pharmaceutics13101537_

Round 1

Reviewer 1 Report

The authors report a simple one-step desolvation method to synthesise gelatin nanoparticles from any types of gelatin regardless of their Bloom number. I have the following suggestions to improve the quality of the manuscript:

1) It would be great if the authors characterise the physicochemical properties of drug-loaded gelatin nanoparticles and provide a comparative table showing the percentage of drug loading content and encapsulation efficiency of the drug-loaded nanoparticles.

 2) The manuscript lacks qualitative analysis of the characterised nanoparticles. In particular, they have provided very low-quality SEM images of the gelatin nanoparticles in Figure 14. It would be better if they use Transmission electron microscopy (TEM) images of the synthesised nanoparticles. Of particular interest is a comparative qualitative image showing the size of the nanoparticles before and after drug loading.

3) The manuscript is very long, with over 25 figures. I suggest making it more concise and add unnecessary data in the supplementary information.

Author Response

We express our great gratitude to the reviewer for comments and thoughtful suggestions. Based on these comments and suggestions, we have made modifications to the original manuscript. We believe that the manuscript has been improved and hope it has reached high journal's standard.

1) It would be great if the authors characterise the physicochemical properties of drug-loaded gelatin nanoparticles and provide a comparative table showing the percentage of drug loading content and encapsulation efficiency of the drug-loaded nanoparticles.

As we mentioned in the text of the article, loading of europium complexes was performed only to demonstrate the principal possibility to encapsulate molecules with poor water solubility by the proposed method. Besides, the loading procedure was not optimized thus we hardly reached the maximum available encapsulation efficiency. Moreover, in the used synthesis conditions we obtained stable nanoparticles only from gelatins B but not from other types of gelatin. Therefore, we suggest that a detailed comparison of loaded and non-loaded nanoparticles will not valuable for the readers. We only performed a size distribution comparison of stable europium-loaded nanoparticles prepared from gelatin B 75 and their empty counterparts. The results can be found in Fig. 9, S13, and S20.

2) The manuscript lacks qualitative analysis of the characterised nanoparticles. In particular, they have provided very low-quality SEM images of the gelatin nanoparticles in Figure 14. It would be better if they use Transmission electron microscopy (TEM) images of the synthesised nanoparticles. Of particular interest is a comparative qualitative image showing the size of the nanoparticles before and after drug loading.

Unfortunately, we are not able to obtain higher-quality images due to the insufficient resolution of our scanning electron microscope. We added transmission electron microscopy image and size distribution graph for nanoparticle loaded and non-loaded with europium complexes. Despite the polymeric nature of nanoparticles did not allow us to reach contrast comparable to that of inorganic particles, we obtained size distributions for both loaded and non-loaded nanoparticles (size distribution histograms can be found in supplementary information).

3) The manuscript is very long, with over 25 figures. I suggest making it more concise and add unnecessary data in the supplementary information.

We shortened the manuscript. Excessive text and figures were transferred to supplementary materials.

Reviewer 2 Report

Manuscript entitled “Modified desolvation method enables simple one-step synthesis of gelatin nanoparticles from different gelatin types with any bloom values” could be interesting for the readers. The manuscript is well-written, structured, and technically sound. However, the manuscript can be improvised on specific points before it is accepted for final publication.

  1. Introduction could be better with more background about the work with up-to-date reference.
  2. Author should write the full form once when mentioning for the first instance.
  3. Authors are advised to add a schematic graphic to show the overall work.
  4. Why results of cell viability assay Fig. 1 & 2 were added in the methodology part? It should be moved to the results and discussion section.
  5. SEM image (Fig. 14) is not clear, add a better quality image. Also, add size distribution.
  6. Fig. 16 F is not clear, revise it. Improve the quality of Figure 25.
  7. There are so many Figures in the text. I think some of them can be moved to the supporting information.
  8. In toxicity assay, if possible, please add the cell viability image.
  9. Conclusion is too long, please make it short and to the point rather than a summary of all the findings.
  10. Also, carefully revise the typos and linguistic errors to make the manuscript error-free.

Author Response

We express our great gratitude to the reviewer for comments and thoughtful suggestions. Based on these comments and suggestions, we have made modifications to the original manuscript. We believe that the manuscript has been  improved and hope it has reached high journal's standard.

  1. Introduction could be better with more background about the work with up-to-date reference.

We do not completely agree with the referee at this point. The introduction consists of about 1000 words and includes 27 references, 50% of which are published between 2019 and 2021. Introduction touches upon the nature and properties of gelatin application of gelatin nanoparticles and contains the key references to articles concerning various aspects and challenges of gelatin nanoparticles synthesis. We think that additional references will be superfluous, especially since the article contains references to most of the newest articles on the research topic.

  1. Author should write the full form once when mentioning for the first instance.

Corrected

  1. Authors are advised to add a schematic graphic to show the overall work.

Novel graphical abstract was prepared

  1. Why results of cell viability assay Fig. 1 & 2 were added in the methodology part? It should be moved to the results and discussion section.

Mentioned figures illustrate methodological aspects of discrimination between live and dead cells by flow cytometry. For clarity, we moved them to the supplementary materials.

  1. SEM image (Fig. 14) is not clear, add a better quality image. Also, add size distribution.

Unfortunately, we are not able to obtain higher-quality images due to the insufficient resolution of our scanning electron microscope. Analysis of nanoparticles in another research center will take too much time. We added a transmission electron microscopy image and size distribution graph for one of the nanoparticle batches.

  1. 16 F is not clear, revise it. Improve the quality of Figure 25.

Both figures were moved to supplementary material and enlarged.

  1. There are so many Figures in the text. I think some of them can be moved to the supporting information.

We moved most of the figures to supplementary material.

  1. In toxicity assay, if possible, please add the cell viability image.

We used the flow cytometry method to assess cell viability. Unfortunately, this method does not allow obtaining of cell images.

  1. Conclusion is too long, please make it short and to the point rather than a summary of all the findings.

We cannot fully agree with the referee, because in conclusion we did not summarize our findings but rather discussed perspectives of further method development (in terms of its scalability, environmental impact, and so on) and pointed limitations of our research.

  1. Also, carefully revise the typos and linguistic errors to make the manuscript error-free.

Corrected

Round 2

Reviewer 1 Report

The revised version can be accepted now.

Author Response

The authors would like to thank the reviewer for his/her comments that helped to improve the manuscript.

Reviewer 2 Report

The paper is much improved and it could be published after minor revision.

Authors are advised to improve the introduction and conclusion based on the reviewer's suggestion.

In the introduction, authors are advised to add more information on the advantage of using gelatin nanoparticles instead of gelatin.

The conclusion should be short rather than a two-page of details about the work.

Author Response

We would like to thank the referee for carefully reading our manuscript and for giving such constructive comments which substantially helped to improve the quality of our paper. The introduction section was modified according to the referee's suggestions. A long text passage with a discussion of the method perspectives and limitations of the study was transferred to the separate section (Section 3.9. Future perspectives and limitations of this study). A short conclusion was also added.